

# Validation of the questionnaire "Pregnancy Vaccine Hesitancy Scale (pVHS)" toward COVID-19 vaccine for Malaysian pregnant women

Nur Azreen Che Mood[1], Zainab Mat Yudin[1,2], Wan Muhamad Amir W Ahmad[1], Azidah Abdul Kadir[2,3], Mohd Noor Norhayati[3], Noorfaizahtul Hanim Md Nawawi[2,3], Erinna Mohamad Zon[2,4] and Norsiah Ali[5]

[1] School of Dental Sciences, Health Campus, Universiti Sains Malaysia, Kubang Kerian, Kelantan, Malaysia
[2] Hospital Universiti Sains Malaysia, Kubang Kerian, Kelantan, Malaysia
[3] Department of Family Medicine, School of Medical Sciences, Health Campus, Universiti Sains Malaysia, Kubang Kerian, Kelantan, Malaysia
[4] Department of Obstetrics and Gynaecology, School of Medical Sciences, Health Campus, Universiti Sains Malaysia, Kubang Kerian, Kelantan, Malaysia
[5] Masjid Tanah Health Clinic, Masjid Tanah, Melaka, Malaysia

## ABSTRACT

**Background**. Pregnancy is one of the risks for severe COVID-19 infection, and receiving a vaccination is one of the effective methods to reduce disease severity. However, COVID-19 vaccine hesitancy among pregnant women remains an issue. This study aims to develop and validate the pregnancy Vaccine Hesitancy Scale (pVHS) toward COVID-19 vaccine for Malaysian pregnant women.

**Method**. An 8-item Malay language pregnancy Vaccine Hesitancy Scale (pVHS-M) for COVID-19 was adapted from the adult Vaccine Hesitancy Scale and validated using Exploratory Factor Analysis. Six expert panels were involved in content validity, and ten pregnant women were involved in face validity. A cross-sectional study on 200 pregnant women was conducted between October 2022 and March 2023 at the Obstetrics and Gynaecology Clinic, Universiti Sains Malaysia, Kelantan.

**Result**. The item-level content validity index is 1.00, demonstrating good relevance of the eight items used to assess COVID-19 vaccine hesitancy. The item-level face validity index obtained is 0.99, indicating that the items were clear and comprehensible. The Cronbach alpha score was 0.944, with factor loadings ranging from 0.79 to 0.89.

**Conclusion**. The pVHS-M demonstrated good internal consistency, indicating that it is a valid and reliable tool for assessing COVID-19 vaccine hesitancy among pregnant women.

Corresponding author
Zainab Mat Yudin, drzainab@usm.my

## INTRODUCTION

Coronavirus disease (COVID-19) is an infectious disease caused by the severe acute respiratory syndrome coronavirus 2 (SARS-CoV-2) and transmitted primarily *via*

respiratory droplets from one person to another when an infected person coughs or sneezes (*Chin, Woon & Chan, 2022*; *Güner, Hasanoğlu & Aktaş, 2020*). Since the coronavirus emergence in late 2019, several mutations such as Alpha, Beta, Delta and Gamma, while the Omicron variant began to be identified in 2021 and continued to evolve, becoming a global threat to particularly high-risk populations, including pregnant women (*Marchildon, 2019*; *Shrestha et al., 2022*; *Tekin et al., 2022*). Vaccination is an effort to give someone a type of vaccine into their body to prevent them from being infected by some disease and eventually reach herd immunity (*Cambridge Dictionary, 2023*; *Hashim et al., 2021*). Before it arrived in the country, the public was advised to practice strict preventive measures such as social distancing and maintaining optimal personal hygiene (*Hashim et al., 2021*). Regarding COVID-19 vaccine side effects, it was possible to feel pain, swelling and redness in the arm where one got injected while also experiencing headache, tiredness, nausea, fever, chills, and muscle pain afterwards (*CDC, 2024*).

Vaccination hesitancy is described as demonstrating a delay in substance acceptance or deliberately refusing to get vaccinated even though immunization could lessen mortality and morbidity (*Pires, 2022*). Globally, the COVID-19 vaccine hesitancy rate ranged from 43.5% to 75.2% (*Lazarus et al., 2022*; *Limbu, Gautam & Pham, 2022*). Global prevalence showed higher vaccine hesitancy among unmarried individuals, women, unemployed, low educational people, households of five or more, older people and college students (*Fajar et al., 2022*; *Lee & You, 2022*). One of the reasons for vaccine hesitancy is the concern about COVID-19 vaccine safety initiated by the rapid development of the COVID-19 vaccine itself (*Machingaidze & Wiysonge, 2021*). Safety issues of the vaccine toward pregnant women become the main barriers to its acceptance (*Boshra et al., 2022*; *Levy et al., 2021*; *Regan et al., 2022*). A systematic review found the COVID-19 vaccine acceptance involving 25,147 pregnant women was around 49% (*Bhattacharya et al., 2022*). A study among East Malaysians found issues of confidence, authority, weakness of mainstream media, complacency, social media, and convenience becoming the factors for COVID-19 vaccine hesitancy (*Jafar et al., 2022*).

A few instruments have been developed to assess knowledge, attitude and practices related to COVID-19 infection and vaccination (*Javier & Garin, 2022*; *Kunno et al., 2022*; *Nemat et al., 2022*; *Regan et al., 2022*; *Takahashi et al., 2022*) however the evaluation of vaccine hesitancy from these instruments were inadequate. The Vaccine Hesitancy Scale has been developed by the WHO SAGE Working Group on Vaccine Hesitancy, which is nonspecific to certain vaccines and age groups and has been utilized in many countries. The Adult Vaccine Hesitancy Scale (aVHS) was modified from its original version to target adult vaccines and provide measures of its reliability and validity relative to influenza vaccine uptake and COVID-19 vaccination acceptance (*Akel et al., 2021*). The present study aimed to perform an adaptation of aVHS for its use among pregnant women in Malaysia.

## MATERIALS & METHODS

The process of adaptation and validation of this questionnaire followed the framework for developing and translating questionnaires for application in medical fields (*Tsang, Royse &*

*Terkawi, 2017*). The permission for aVHS adaptation and modification was obtained from the author (*Akel et al., 2021*).

## Stage 1: translation and adaptation

Forward and backward translation of aVHS in Malay language was conducted. Two family medicine specialists who were proficient in English and familiar with the topic individually translated the aVHS into the Malay version. Each translation was examined and combined into one Malay translation. Two more bilingual translators who were blind to the original English version then translated the content back into English. The differences that were found were examined and improved so that the Malay translation accurately conveyed the original aVHS questionnaire's meaning. Next, it underwent modification by expert members involving three family medicine specialists, one public health specialist, and one obstetrician to obtain the modified version (M1). The M1 consisted of 10 items (items pL1 to pL10). We made all items positively worded, including the pL10. The choice of Likert-type scale continuum was modified into 1; strongly disagree, 2; disagree, 3; not sure, 4; agree to 5; strongly agree to represent the least number as the more hesitant. Two items (pL5 and pL9) were given reverse scoring. Following this, an assessment of the content validity involving two family medicine specialists, one public health specialist, one obstetrician, one nursing tutor, and one public health nurse was performed to produce the M2. Panelists rated the items on a Likert scale for content relevance and clarity. For each item, relevancy was rated from 1 (the item was very irrelevant) to 4 (the item is very relevant), while clarity was rated from 1 (the item is totally not clear) to 4 (the item is very clear). Then, M2 underwent face validity among ten pregnant women to identify any issues in interpreting the items, and no problems were found. In the absence of modifications, the final version (M2) is utilized for the validation stage and named pVHS-M.

## Stage 2: validation

The study sample was a convenience sample of 200 pregnant women between October 2022 and March 2023 at the Obstetrics and Gynaecology Clinic, Hospital Universiti Sains Malaysia (HUSM), Kelantan. A trained researcher conducted the data collection. Pregnant women were approached at the clinic waiting area, and screening for eligibility was performed. The inclusion criteria include 18 years and above with no underlying acute illness or acute condition during the day of questionnaire administration. Eligible pregnant women were invited to participate in the study. The questionnaire was completed individually, but the participants were able to ask a researcher for help in case of need. The approximate time to complete the questionnaire was 15 to 20 min.

The study was approved by the Human Research Ethics Committee (JEPEM) USM USM/JEPeM/22050297 and Medical Research & Ethic Committee (MREC), Ministry of Health Malaysia (22-01456-AQA). All subjects provided written informed consent.

## Data analysis

The SPSS software version 26.0 was used for data analysis. We used descriptive analysis to describe the sociodemographic characteristics. Mean and SD were used to describe

**Table 1 Sociodemographic characteristics of the participants (n = 200).**

| Variables | n (%) | Mean (SD) |
|---|---|---|
| Age (years) | | 30.5 (5.51) |
| Educational level | | |
| Primary education and below | 4 (2.0) | |
| Secondary education | 94 (47.0) | |
| Tertiary education | 102 (51.0) | |
| Occupation | | |
| Housewives | 102 (51.0) | |
| Government worker | 43 (21.5) | |
| Non-Government worker | 27 (13.5) | |
| Self-employed and others | 28 (14.0) | |

Notes.
Primary education and below: No education, Standard 6; Secondary education: SRP/PT3/SPM; Tertiary education: Diploma/Advanced Diploma, Bachelor's degree, Master's degree or higher.

continuous variables (age), and frequency and proportion were used to describe the categorical variables (education and occupation). The Exploratory Factor Analysis (EFA) was conducted to test the construct validity by assessing the strength and direction of the relationship between each item and the underlying factor (*Cronbach & Meehl, 1955*). High factor loadings (close to 1) suggest that an item is strongly associated with the factor, providing evidence that the items are measuring the construct. The optimal number of factors and rational items or indicators for the factors was determined (*Koyuncu & Kılıç, 2019*) using the factor loading, suitable factor rotation method and significance values for Kaiser-Meyer-Olkin (KMO) and Barlett tests.

# RESULTS

The psychometric evaluation participants comprised 200 pregnant women with a mean age of 30.5 years and half of them had tertiary education. Sociodemographic data are detailed in Table 1. The items of the newly validated COVID-19 vaccine hesitancy scale (pVHS-M) underwent review and modification by expert panels as detailed in Table 2. Items pL1 to pL9 were modified accordingly to tally with the COVID-19 vaccine and item pL10 was positively worded. A content validation form with all items (pL1 to pL10) of the questionnaire was given to six-panel experts to be rated on the relevancy and provided feedback on the topics. Item-level content validity index (I-CVI) using scale-level content validity index universal agreement method (S-CVI/UA) = 1.0 while scale-level content validity index by averaging method (S-CVI /Ave) = 1.0. The average face validity index (FVI) involving 10 respondents was 0.99.

## Validation of pVHS-M

A total of ten items were used to conduct EFA by using the principal component axis as the extraction method. Bartlett's test of sphericity showed that the results were significant with $p < 0.001$. The KMO measure of sampling adequacy yielded a value of 0.886, indicating that the sample size was large enough to assess the factor structure. The value was above

**Table 2  The items for COVID−19 vaccine hesitancy among pregnant women (p-VHS-M).**

| Items | Description |
|---|---|
| pL1 | COVID-19 vaccine is important for my health. |
| | *Vaksin COVID-19 adalah penting untuk kesihatan saya.* |
| pL2 | COVID-19 vaccine is effective to reduce severity of infection. |
| | *Vaksin COVID-19 adalah berkesan untuk mengurangkan keterukan jangkitan.* |
| pL3 | COVID-19 vaccination is important for the health of others in my community. |
| | *Vaksinasi COVID-19 adalah penting untuk kesihatan masyarakat di komuniti saya.* |
| pL4 | COVID-19 vaccine recommended by the Ministry of Health Malaysia (KMM) are beneficial. |
| | *Vaksin COVID-19 yang disarankan oleh kerajaan Malaysia adalah bermanfaat.* |
| pL5 | COVID-19 vaccine is a new vaccine which carry more side effect. |
| | *Vaksin COVID-19 adalah vaksin baru yang memberi lebih banyak kesan sampingan.* |
| pL6 | The information I receive about COVID-19 vaccine from the Ministry of health Malaysia (Malaysia MoH) is trustworthy. |
| | *Maklumat yang saya terima mengenai vaksin COVID-19 dari Kementerian Kesihatan Malaysia (KKM) boleh dipercayai.* |
| pL7 | Getting COVID-19 vaccination is a good way to protect me from severe disease infection. |
| | *Menerima suntikan vaksin COVID-19 adalah kaedah yang bagus untuk melindungi saya dari jangkitan penyakit yang teruk.* |
| pL8 | Generally, I agree to all recommendation of Malaysia Ministry of Health about the COVID-19 Vaccine. |
| | *Secara amnya, saya bersetuju dengan semua saranan Kementerian Kesihatan Malaysia berkaitan vaksin COVID-19.* |
| pL9 | I am concerned about serious adverse effects of vaccine. |
| | *Saya bimbang terhadap kesan sampingan vaksin yang serius.* |
| pL10 | I do need COVID-19 vaccine as a preventive measure against new variants. |
| | *Saya memerlukan vaksin COVID-19 sebagai langkah pencegahan menghadapi varian baru.* |

0.7, so the data were sufficient to proceed with the factor analysis. EFA had identified and explained the patterns of correlations between the observed items. Table 3 shows the communality and factor loadings on the first EFA on ten items of pVHS-M. Sufficient convergent validity was demonstrated by the factors as eight items of the questionnaire had their factor loadings above the recommended threshold of 0.30 for 200 sample size. The acceptable factor loading cut off point of this study was 0.3, hence eight items with factor loadings above 0.3 were maintained. A high Cronbach's alpha (typically above 0.7) suggests good internal consistency (*Hajjar, 2018*; *Taber, 2018*; *Whitley Jr & Kite, 2012*). Meanwhile the other two items had low factor loadings, with 0.096 and 0.119 for item pL5 and item pL9 consecutively. After consideration, these two items were removed, and EFA was conducted again on the remaining eight items.

However, the original item 5 of an aVHS (New vaccines carry more risks than older vaccines.) and the original item 10 of an aVHS (I do not need vaccines for diseases that are no longer common) were found confusing by the experts. Therefore, these items were revised and modified accordingly to a simpler Malay language structure as per content expert recommendation and turned into item pL5 (COVID-19 vaccine is a new vaccine which carry more side effect) and item pL10 (I do need COVID-19 vaccine as a preventive measure against new variants). From the ten items of pVHS-M, it was found that the factor loading of eight items was adequate, ranging from 0.79 to 0.89, while two items had

**Table 3   Exploratory factor analysis on 10-item of COVID-19 vaccine hesitancy among pregnant women (p-VHS-M).**

| Item | Communalities | Factor loading | Cronbach |
|------|---------------|----------------|----------|
| pL1 | 0.629 | 0.793 | |
| pL2 | 0.796 | 0.892 | |
| pL3 | 0.781 | 0.884 | |
| pL4 | 0.783 | 0.885 | |
| pL5 | *0.009 | *0.096 | 0.859 |
| pL6 | 0.567 | 0.753 | |
| pL7 | 0.638 | 0.798 | |
| pL8 | 0.711 | 0.843 | |
| pL9 | *0.014 | *0.119 | |
| pL10 | 0.571 | 0.755 | |

Notes.

*The items which low value and indicate weaker associations or possible overlap with other constructs.

**Table 4   Exploratory factor analysis on 8-items of p-VHS-M.**

| Item | Communalities | Factor loading | Cronbach |
|------|---------------|----------------|----------|
| pL1 | .629 | .793 | |
| pL2 | .796 | .891 | |
| pL3 | .781 | .884 | |
| pL4 | .783 | .886 | 0.944 |
| pL5 | .567 | .753 | |
| pL6 | .638 | .799 | |
| pL7 | .711 | .843 | |
| pL8 | .571 | .757 | |

low factor loadings: less than 0.2 while the recommended and acceptable factor loadings should be higher than 0.6 (*Awang et al., 2015*). The low factor loadings were 0.096 and 0.119 for item pL5 and item pL9 consecutively. These two items were removed, and EFA was conducted again on the remaining eight items. Results for the second round of EFA were excellent since all eight items had high factor loadings.

## Reliability

Internal consistency reliability was used to assess the extent to which the items within a questionnaire were interrelated. The test and retest reliability were conducted and improved from 0.859 to 0. 944 (Table 4). The final Cronbach's Alpha coefficient of reliability test produced values of 0.94, which means the tool is reliable in assessing vaccine hesitancy among pregnant women. As a conclusion of EFA, two items were removed; thus, eight items were retained in pVHS-M.

## DISCUSSION

Vaccine hesitancy toward COVID-19 vaccination among pregnant women is a threat to maternal morbidity and mortality in pregnancy, and worldwide including Malaysia. It has demonstrated an increment in deaths among pregnant women during the delta strain wave of COVID-19. The aVHS provide an adequate validity and reliability during cross-cultural adaptation among adult population in various countries including Italy, Arab and Turkey (*Çelik, Özer & Zincir, 2023*; *ElHafeez et al., 2021*; *Ledda et al., 2022*).

The pVHS-M was developed from an adaptation of the aVHS. In a situation where a suitable questionnaire is unavailable, it is crucial to design a questionnaire to collect data that can produce a generalizable and interpretable result (*Jenn, 2006*). The adaptation of a questionnaire is acceptable, and even though there were several methods for the process, there were no specific guidelines for this process (*Epstein, Santo & Guillemin, 2015*). The aVHS is a well-designed assessment tool to assess vaccine hesitancy among adults. This tool used a Likert scale with five fixed response choices (*Bowling, 2002*). The content and face validation, as well as exploratory factor analysis for pVHS-M were conducted on the unidimensional domain. Its content validity was established through two methods, qualitative and quantitative (*McKenzie et al., 1999*). The number of experts for content validation should be at least six and not exceed ten (*Yusoff, 2019a*). For the qualitative content validation, the experts critically reviewed the domain and its items before scoring each item. The experts provided verbal or written comments to improve the relevance and clarity of items to the targeted domain. All comments were taken into consideration to refine the domains and their items. Meanwhile, a CVI was used to provide quantitative evidence of the content's validity (*Yusoff, 2019a*). The evaluation of items was based on the item's relevancy and clarity. Items were provided in Malay language.

The scale's face validity was established through two methods: qualitative and quantitative approaches (*Yusoff, 2019b*). The minimum sample for face validity should be three for a sample size of six to 20; thus, for a 200-sample size, there should be at least 30 pregnant women for face validity (*Yusoff, 2019b*). However, a minimum of ten pregnant women were interviewed to complete the scale.

This questionnaire obtained a 0.99 face validity index, which can be indicated as clarified and comprehensible among the ten panels since it is recommended that an item face validation index (I-FVI) value above 0.8 can be acceptable for inter-rater agreement in the questionnaire (*Hadie et al., 2017*; *Yusoff, 2019b*). Internal validation for the pVHS-M structure obtained excellent value for its vaccine hesitancy domain. The values indicated the remaining eight items are reliable for assessing vaccine hesitancy among pregnant women in Malaysia.

This study has several strengths, covering the parts of the research. First, the topic of COVID-19 is relatively recent; thus, it would gain as much focus, especially regarding vaccination, and provide a tool to assess such behavior of vaccine hesitance. It is a step towards a solution, and thus, it would assist more COVID-19 research initiatives in the future. Other than that, the population involved in the study is among the high-risk population, which would need a special encounter when facing these pandemic outbreaks.

It would be simpler to evaluate and be aware of their mental state and vaccination hesitancy rate to carry out the preventive intervention for them. Therefore, paying extra attention to the high-risk population should be a good step in this study. Since the study was conducted among pregnant women in Kelantan, the majority are Malay and Muslim. This was one of the limitations. The subject population was from a tertiary hospital antenatal clinic, where most of the pregnant women have high-risk pregnancies. More research is required to assess the potential benefit of the pVHS-M in primary care settings, particularly in anticipating the vaccine acceptance before the subsequent COVID-19 variant outbreak.

## CONCLUSION

The pVHS-M demonstrates a good and reliable psychometric property through the validation process. Moreover, it is a valid and reliable tool to assess the COVID-19 vaccine hesitancy among pregnant women.

### Funding

This study received a Fundamental Research Grant Scheme (FRGS) from the Ministry of Education (MOE) (FRGS/1/2022/SKK06/USM/02/6). The funders had no role in study design, data collection and analysis, decision to publish, or preparation of the manuscript.

### Grant Disclosures

The following grant information was disclosed by the authors:
Ministry of Education (MOE): FRGS/1/2022/SKK06/USM/02/6.

### Competing Interests

Norhayati Mohd Noor is an Academic Editor for PeerJ. The authors declare that they have no other competing interests.

### Author Contributions

- Nur Azreen Che Mood performed the experiments, analyzed the data, prepared figures and/or tables, authored or reviewed drafts of the article, and approved the final draft.
- Zainab Mat Yudin conceived and designed the experiments, performed the experiments, analyzed the data, prepared figures and/or tables, authored or reviewed drafts of the article, and approved the final draft.
- Wan Muhamad Amir W Ahmad analyzed the data, prepared figures and/or tables, authored or reviewed drafts of the article, and approved the final draft.
- Azidah Abdul Kadir conceived and designed the experiments, performed the experiments, authored or reviewed drafts of the article, and approved the final draft.
- Mohd Noor Norhayati conceived and designed the experiments, performed the experiments, authored or reviewed drafts of the article, and approved the final draft.
- Noorfaizahtul Hanim Md Nawawi conceived and designed the experiments, performed the experiments, prepared figures and/or tables, authored or reviewed drafts of the article, and approved the final draft.

- Erinna Mohamad Zon conceived and designed the experiments, authored or reviewed drafts of the article, reviewed the methodology, and approved the final draft.
- Norsiah Ali conceived and designed the experiments, authored or reviewed drafts of the article, reviewed the methodology, and approved the final draft.

## Human Ethics

The following information was supplied relating to ethical approvals (i.e., approving body and any reference numbers):

Human Research Ethics Committee (JEPEM) USM USM/JEPeM/22050297 and Medical Research & Ethic Committee (MREC), Ministry of Health Malaysia (22-01456-AQA).

## Data Availability

The raw data, Malay language scale and user guide&code are available in the Supplementary Files 1, 2 and 3.

## Supplemental Information

Supplemental information for this article can be found online at http://dx.doi.org/10.7717/peerj.17134#supplemental-information.

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
