# Peer review of "Validation of the questionnaire "Pregnancy Vaccine Hesitancy Scale (pVHS)" toward COVID-19 vaccine for Malaysian pregnant women"

_PeerJ, doi:10.7717/peerj.17134_

## Round 0.1 · original submission · Minor Revisions

Dear authors, thank you for your patience. Please, address the reviewers' comments.

Reviewer 1 ·

Basic reporting

Overall, the manuscript is written in clear language, but minor grammar errors were detected. For example, line 7 should be “one of the effective methods”. Lines 21-22, lines 47-54, and some other places are a bit confusing. Sufficient references and background are provided. Figures and tables included are relevant to the hypothesis. The English language should be improved to ensure that an international audience can clearly understand your text. Some examples where the language could be improved:

Experimental design

- Lines 116-117 should be more specific, for example, mean and SD were used to describe continuous variables, and frequency and proportion were used to describe categorical variables.
- Line 106-108 should go to results and data analysis.
- Lines 160-162 are a bit confusing. Please rephrase the sentences to show that the two items with low factor loadings were from the original 10 items, not from the 8 remaining items.

Validity of the findings

- Please include some of the potential study limitations in the discussion section.

Reviewer 2 ·

Basic reporting

Generally acceptably written, well organized, interesting topic
References seem fine, might be missing a few relevant ones (there seem to be very similar studies in terms of methods utilized, albeit in other populations, that I feel should be cited; if this work was inspired by other authors work, would recommend citing those); citations are in odd format (not numbered?)
Data shared in supplemental
Results are relevant to hypothesis

Experimental design

Clear research inquiry, relevant, meaningful
Fulfills very specific knowledge gap
High technical component of investigation
No ethical concerns
Methods reproducible

Validity of the findings

Robust relevant statistical parameters were reported
Good results
Quality thresholds are clear, important for results interpretation

Additional comments

Limitations can be more explicitly/formally discussed
would try to eliminate subjective words, such as 'extremely important' and similar.
Don't need introductory words in sentences, "moreover", "therefore", etc
Double check grammar and phrasings

---

## Round 0.2 · accepted · Accept

Dear authors, I am happy that your manuscript has now been accepted for publication. Thank you for submitting to PeerJ and for sharing your important work.

Reviewer 1 ·

Basic reporting

The authors have adequately addressed my comments. Therefore, I have no further comments.

Experimental design

The authors have adequately addressed my comments. Therefore, I have no further comments.

Validity of the findings

The authors have adequately addressed my comments. Therefore, I have no further comments.

Reviewer 2 ·

Basic reporting

Improved, concerns addresssed

Experimental design

Improved, concerns addresssed

Validity of the findings

Improved, concerns addresssed

Additional comments

Improved, concerns addresssed